# Immune Response Dysfunction in Chronic Lymphocytic Leukemia: Dissecting Molecular Mechanisms and Microenvironmental Conditions

**DOI:** 10.3390/ijms21051825

**Published:** 2020-03-06

**Authors:** Francesca Arruga, Benjamin Baffour Gyau, Andrea Iannello, Nicoletta Vitale, Tiziana Vaisitti, Silvia Deaglio

**Affiliations:** Department of Medical Sciences, University of Turin, via Nizza, 52, 10126 Torino, Italy; francesca.arruga@unito.it (F.A.); benjaminbaffour.gyau@unito.it (B.B.G.); andrea.iannello@unito.it (A.I.); nicoletta.vitale@unito.it (N.V.); tiziana.vaisitti@unito.it (T.V.)

**Keywords:** immunosuppression, tolerance, immune checkpoints, microenvironment, hypoxia

## Abstract

Representing the major cause of morbidity and mortality for chronic lymphocytic leukemia (CLL) patients, immunosuppression is a common feature of the disease. Effectors of the innate and the adaptive immune response show marked dysfunction and skewing towards the generation of a tolerant environment that favors disease expansion. Major deregulations are found in the T lymphocyte compartment, with inhibition of CD8^+^ cytotoxic and CD4^+^ activated effector T cells, replaced by exhausted and more tolerogenic subsets. Likewise, differentiation of monocytes towards a suppressive M2-like phenotype is induced at the expense of pro-inflammatory sub-populations. Thanks to their B-regulatory phenotype, leukemic cells play a central role in driving immunosuppression, progressively inhibiting immune responses. A number of signaling cascades triggered by soluble mediators and cell–cell contacts contribute to immunomodulation in CLL, fostered also by local environmental conditions, such as hypoxia and derived metabolic acidosis. Specifically, molecular pathways modulating T-cell activity in CLL, spanning from the best known cytotoxic T lymphocyte antigen-4 (CTLA-4) and programmed cell death 1 (PD-1) to the emerging T cell immunoreceptor with immunoglobulin and immunoreceptor tyrosine-based inhibition motif domains (TIGIT)/CD155 axes, are attracting increasing research interest and therapeutic relevance also in the CLL field. On the other hand, in the microenvironment, the B cell receptor (BCR), which is undoubtedly the master regulator of leukemic cell behavior, plays an important role in orchestrating immune responses, as well. Lastly, local conditions of hypoxia, typical of the lymphoid niche, have major effects both on CLL cells and on non-leukemic immune cells, partly mediated through adenosine signaling, for which novel specific inhibitors are currently under development. In summary, this review will provide an overview of the molecular and microenvironmental mechanisms that modify innate and adaptive immune responses of CLL patients, focusing attention on those that may have therapeutic implications.

## 1. Introduction

Chronic lymphocytic leukemia (CLL) is a blood malignancy of mature B cells, which clonally expand and accumulate in the peripheral blood, bone marrow, lymph nodes and spleen [1]. Disease progression and outcome are defined both by a genetic component, which includes a relatively wide spectrum of genetic lesions useful for risk stratification, and by significant contributions from microenvironmental interactions providing signals that influence leukemic cell behavior [2]. Despite being a clinically and molecularly heterogeneous disease, CLL is typically characterized by significant perturbations of the immune system, involving both innate and adaptive immune responses and leading to immune suppression from the early stages. Dysfunction of the immune system in turn increases the incidence of secondary malignancies and infections, which represent the major cause of morbidity and mortality for CLL patients [3,4]. Immunosuppression and increased risk for infections can be due to patient-, treatment- and disease-related factors. Advanced age is a known risk factor for infections in patients with hematological malignancies in general and it certainly plays a role in CLL where the majority of patients are elderly subjects. The presence of comorbidities and the overall fitness status of the patients also influence the susceptibility to infections [5]. Type, duration, number and combinations of therapies importantly contribute to immune suppression in most patients and all treatment-related factors are in close correlation with individual features. As an example, a number of studies observed that therapy for relapsed disease is associated with an increased risk of infections compared to the same treatment used as front-line therapy, likely because of disease evolution and decreased patients’ fitness [4,6]. Lastly, despite immunosuppression being a condition already observed in early CLL stages, advanced/relapsed disease or poor response to therapy represent disease-related factors associated with a more severe dysfunction of the immune system [7]. The picture arising from these observations suggests that the classification in patient-, treatment- or disease-related factors influencing immune responses may sound oversimplified and that the combination of different conditions leads to a dynamic and self-fostering dysregulation of the immune system.

Quantitative and qualitative defects are observed in almost all CLL patients starting from diagnosis (summarized in Figure 1). Alterations of the innate immune system include defective function of neutrophils, natural killer (NK) cells and decreased complement activity. On the side of the adaptive immune response, deficits in cell-mediated immunity with hypogammaglobulinemia, down-regulation of T-cell function and defects in antibody dependent cellular cytotoxicity were reported [8].

Among immune system dysfunctions observed in CLL, it is worthwhile mentioning that a significant proportion of patients experiences autoimmunity-related clinical issues. The relationship between CLL and autoimmune cytopenias, particularly autoimmune hemolytic anemia and immune thrombocytopenia, is well established. The responsible mechanism is ascribed to the fact that the global scenario of immunosuppression that characterizes CLL favors the expansion of autoreactive leukemic cells [9].

Growing evidence indicates that CLL cells modulate phenotype and functions of immune cells from the innate and adaptive immune system through a number of surface molecules and soluble factors. This review will navigate the main immune system alterations observed in CLL, exploring the molecular mechanisms and cell interactions responsible for skewing immune cell phenotypes and functions towards tolerance.

## 2. Innate Immune Response Dysfunction in Chronic Lymphocytic Leukemia (CLL)

Alterations of multiple elements and effectors of the innate immune response are found in CLL patients (Figure 1). One of the most characteristic features of CLL innate immune response is the presence of a macrophage population with pro-tumoral (M2-like) phenotype, termed nurse-like cells (NLCs). These cells were originally described as spontaneously differentiating in vitro from circulating monocytes [10]. It is now clear that skewing of macrophage populations towards tolerance is primarily driven by CLL cells themselves, through the secretion of soluble factors [e.g., IL-10), adenosine, nicotinamide phosphorybosyltransferase (NAMPT)], as no M2 differentiation of CD14^+^ monocytes is achieved when cultured in vitro with normal B cells [11]. In fact, an overall increase of circulating monocytes count was described in CLL patients, but such cells show a gene expression profile associated with immunosuppressive properties [12]. Consistently, a number of in vitro studies showed that NLCs are functionally equivalent to tumor-associated macrophages (TAMs) described in solid tumors, and express CD163, CD206, CD14 and CD68 on the cell surface [13]. Deficiencies in β-glucuronidase, lysozyme and myeloperoxidase were reported resulting in a relatively refractory state and impaired response to classic pathogens [14]. In addition to the lack of pro-inflammatory responses, NLCs actively support CLL cell survival in vitro and protect them from spontaneous and drug-induced apoptosis through multiple membrane-bound and soluble factors. Among others, the CXCL12/CXCR4 axis is one of the most studied pathways through which NLCs influence CLL cell behavior and survival [10]. In addition, NLCs also express molecules such as BAFF (B-cell activating factor), APRIL (a proliferation inducing ligand), CD31 and Plexin B1, which contribute to proliferation and survival of leukemic cells [15,16]. Furthermore, NLCs are attracted by CLL cells that secrete CX3CL1/Fractalkine and, once in contact, adhesion of NLCs and CLL cells may be strengthened by homotypic (CLL to CLL or NLC to NLC) or heterotypic (CLL to NLC) interactions driven by surface CX3CL1 and CX3CR1 [17]. 

Dendritic cells (DCs) are also dysfunctional in CLL, with alteration of the cytokines profile, lack of the maturation antigen CD83 and the co-stimulatory molecule CD80, and inability to activate proper type 1 T cell responses [18]. The mechanism behind this phenotype is believed to involve, among others, the upregulation of the negative regulator SOCS5 that prevents STAT6 activation in response to IL-4R signaling, thus impairing differentiation of functionally mature DCs and leading to reduced secretion of pro-inflammatory cytokines [19].

Among myeloid cells, myeloid-derived suppressor cells (MDSCs) represent a heterogeneous population of progenitors and precursors of granulocytes, macrophages, and dendritic cells. These cells were identified as a major component of the microenvironment in various malignancies including CLL and can inhibit T-cell responses limiting immune therapeutic approaches. These cells are defined as a CD14^+^HLA-DR^lo^ population and it was observed that an increased frequency of this cell subset is associated with poor prognosis, higher leukemic burden and decreased time to progression in CLL patients [20]. Few years ago, Jitschin and colleagues showed that CLL cells and MDSC can influence one another behavior and differentiation by establishing a bidirectional connection. On the one side, CLL cells directly induce the expansion of a subset of CD14^+^HLA-DR^lo^ MDSCs expressing high amounts of indoleamine-2,3-dioxygenase (IDO). These cells are increased in CLL patients, even at the earliest stages, and suppress both T-cell activation and proliferation. On the other side, MDSC enhance the immunosuppressive phenotype of leukemic cells and promote a CLL-driven differentiation of regulatory T cells. All these effects depend on IDO expression as they were lost in the presence of neutralizing antibodies against IDO, providing also an interesting therapeutic rationale [21].

Circulating NK cells were found increased in CLL patients. However, these cells show elevated CD27 expression, which is normally associated with a decline in mature cells, and defective expression of the NKG2D coreceptor [22]. Functionally, NK cells from CLL patients show reduced degranulation responses toward transformed B cells and appear to be more sensitive to activation-induced cell death. Specifically, NK cells expressing the inhibitory killer cell Ig-like receptors (KIR)2DL1 and/or KIR3DL1 are markedly reduced in frequency and viability, with progressive loss of function over disease course. These observations suggest that mature KIR-expressing NK cells can respond against circulating malignant B cells, but undergo activation-induced apoptosis favoring the expansion of non-functional NK cells [23].

Similarly, the percentage of neutrophils is higher in CLL patients compared to healthy subjects, despite severe functional defects likely due to myeloperoxidase deficiency and impaired migratory and phagocytic abilities. Phenotypically, neutrophils from CLL patients upregulate surface marker of activation, such as CD54, irrespective of infections, probably as a result of a chronic stimulation of the immune system. In contrast, downmodulation of surface Toll-like receptor 2 (TLR-2) indicates the acquisition of a tolerant state, in line with the overall immunosuppression. Furthermore, CLL neutrophils produce increased amounts of reactive oxygen species (ROS), likely affecting cytotoxic activity of T cells and NK cells, and contributing to the generation of a genotoxic environment, fostering genetic instability and disease progression [24].

Lastly, reduced levels of complement proteins, including some of the C1-C4 components, were described in a significant proportion of CLL patients, likely contributing to increased risk of infection as a consequence of impaired pathogens coating [25].

## 3. Adaptive Immune Response Dysfunction in CLL

CLL is a disorder of B-lymphocytes and it is therefore intrinsically characterized by adaptive immune response dysfunction (Figure 1). Circulating CLL cells share phenotypic features of regulatory B-cells (Bregs), such as surface expression of CD5, CD24 and CD27. Bregs negatively regulate immune cell responses mostly by secreting IL-10. IL-10 production is boosted by a range of stimuli in vitro, including T-cell co-stimulation via CD40L or IL-4 or BAFF of the TNF family from monocytes/macrophages, and is remarkably increased in vivo in CLL lymph nodes. This in turn affects different features of the adaptive immune response, with impairment of T-cell activity and the increase of regulatory T cells (Tregs), but also of the innate immune response, with stimulation of inhibitory NK cells, decrease of monocyte- and DC-generated inflammatory cytokines, and suppression of macrophage functions [26]. Immune suppression in CLL is also a consequence of overall hypogammaglobulinemia, resulting from decreased survival and functions of plasmacells, whose severity may vary and increase with disease progression. In early-stage CLL, hypogammaglobulinemia generally involves only one serum immunoglobulin (Ig), while in advanced CLL it becomes a condition affecting all Ig classes [8,27].

Alterations of the T cell repertoire are already present at early stages and worsen with disease progression. Interestingly, the absolute count of CD8^+^ T lymphocytes is increased in early-stage CLL, unbalancing the CD4^+^/CD8^+^ ratio, likely as a result of a tentative CLL-targeting adaptive immune response [28]. Despite increased numbers, circulating T cells in CLL show an abnormal phenotype, with increased expression of exhaustion markers such as cytotoxic T lymphocyte antigen-4 (CTLA-4), lymphocyte activation gene 3 (LAG-3), CD244, CD160 and PD-1. These phenotypic changes also correlate with marked functional defects and significantly reduced cytotoxic capacity and proliferative ability [29]. Exhaustion of CLL T cells is different from virus-induced exhaustion in that they retain the ability to secrete cytokines such as TNF-α that protects CLL cells from apoptosis and induces their proliferation, suggesting that CLL T cells may have been co-opted to shape a pro-leukemic microenvironment [30]. CLL cells manipulate T lymphocytes to gain a survival advantage by turning off cytotoxic T cell function, likely as a consequence of defective linker for activation (LAT) of T cells. Leukemic B cells form a dysfunctional immune synapse with cytotoxic T lymphocytes (CTLs), which in turn exhibit impaired packaging of the granzyme into vesicles and ineffective degranulation, thereby favoring CLL break-out from CTL-mediated cytotoxicity [31]. Physiologically, T cell activation at the immune synapse is tightly controlled by co-stimulatory and co-inhibitory molecules that synergize with the T cell receptor (TCR) to promote (e.g., CD28) or constrain (e.g., CTLA-4) T cell activation and functions [32]. Similar to other tumor models, CLL cells take advantage and promote the upregulation of a regulatory feedback inhibition, including among other the PD-L1/PD-1 axis, to evade immune-attack and maintain tolerance.

In addition to direct CLL-T cell inhibitory interactions, there is also a third-party immunomodulatory effect mediated by the expansion of CD4^+^/CD25^hi^/FOXP3+/CD127dim^/−^ Tregs in the CD4^+^ subset, which contribute to suppressing host immune response by targeting other immune cells functions [33,34]. In physiologic conditions, Tregs are essential for maintaining tolerance and limit auto-immune disease, but their expansion in neoplastic conditions, including CLL, is generally associated with unfavorable prognosis. Functionally, Tregs exert an inhibitory effect on conventional CD4^+^ on cytotoxic CD8^+^ T cells and on NK cells, mainly through the secretion of suppressive cytokines, such as TGF-β and IL-10 [35]. Tregs generation in CLL is likely promoted by local conditions in the lymph node microenvironment (e.g., hypoxia) and by co-stimulatory signals coming directly from CLL cells, such as CD27/CD70 or B7/CTLA-4 interactions. Their expansion in CLL is further facilitated by the observed Bcl2 upregulation that favors Tregs apoptosis resistance [36,37]. The importance of reciprocal interactions occurring between CLL cells and T lymphocytes in generating a favorable environment, is also witnessed by in vivo evidence where the presence of CD4^+^ T cells is required to allow engraftment of leukemic B cells in patient-derived xenograft models [38].

The following paragraphs will describe the major immunomodulatory pathways, molecules or conditions that over the years have acquired increasing relevance in the landscape of immune response perturbations in CLL.

## 4. Molecular Mechanisms Driving Immune Response Dysfunction in CLL

### 4.1. Cytotoxic T Lymphocyte Antigen-4 (CTLA-4)/B7

CTLA-4 (CD152), expressed on T cells, belongs to the network of immune checkpoint signaling modulating T cell activity and was the first co-inhibitory molecule identified. Its engagement, upon TCR activation, downmodulates the amplitude of T cell responses through the inhibition of co-stimulatory signals coming from CD28 (Figure 2) [39]. CTLA-4 represents the paradigm for regulatory feedback inhibition, as its transport to the T cell surface is initiated by activation of CD28 itself: the stronger the CD28-mediated signal, the greater the amount of CTLA-4 that is trafficked to the T-cell surface. CTLA-4 competes with CD28 for binding to shared ligands CD80 (B7.1) and CD86 (B7.2), thus counteracting T cell activation once antigen recognition occurs [40]. Once bound, the CTLA-4–CD80/CD86 complex undergoes trans-endocytosis [41] and activates PTPN11 phosphatase to inhibit internal activation signaling of the TCR (counteracting both tyrosine and serine/threonine kinase signals induced by TCR and CD28) [39]. Trans-endocytosis of the complex mediates T-cell inhibition also through removal of CD80 and CD86, thus preventing CD28 ligation and leading to a signaling-independent inhibition [41].

The major immunoregulatory activity of CTLA-4 relies on the down-modulation of CD4^+^ T cell activation, being predominately expressed on CD4^+^ helper T cells, although a direct inhibitory effect on CD8^+^ T cells was reported [42]. CTLA-4 is also constitutively expressed by Tregs where it contributes to enhance their regulatory activity and suppressive functions [43]. CD4^+^, CD8^+^ and regulatory T cells from CLL patients express increased amounts of surface and cytoplasmic CTLA-4 compared to age- and sex-matched healthy subjects [44]. CTLA-4 expression on T cells in CLL correlated with advanced Rai stage, unfavorable cytogenetics, unmutated IGHV status and Zap-70 expression in leukemic cells. A direct association between surface CTLA-4 and increased proportion of Tregs or low serum IgG and IgA levels was also observed, confirming a scenario of immune dysfunction [45].

CTLA-4 suppression of anti-CLL immunity primarily resides in the microenvironment of the secondary lymphoid organs, where T cell activation occurs. Given the tight interactions between T lymphocytes and leukemic B cells taking place in the niche, and their importance in CLL biology and outcome, CTLA-4 blockade could represent a winning strategy by boosting effector T-cell helper activity and humoral immunity, while inhibiting the activity of Tregs [43]. However, compared to solid tumors, where anti-CTLA-4 blocking antibody ipilimumab is widely adopted with significant benefits for patients [46], this approach has only recently entered the clinical practice for hematological disorders and few trials are currently ongoing in CLL (NCT01822509, NCT00089076; www.clinicaltrials.gov) [47].

CTLA-4 was also found to be expressed by CLL cells themselves, although its biological implications remain controversial. Kosmaczewska et al. observed increased CTLA-4 expression both on the cell surface and in the cytoplasm on CLL cells, again associated with adverse prognosis and disease progression. The authors suggest a possible correlation between CTLA-4 expression and a status of partial activation of B leukemic cells [48]. In contrast, Mittal and colleagues described that CTLA-4 expression on CLL cells is generally associated with good clinical outcome and that the presence of a polymorphism of *CTLA-4* is correlated to increased risk and advanced Rai stages in CLL [49]. They showed that in vitro CTLA-4 down-modulation increases the proliferation rate of leukemic cells and upregulates surface expression of CD38, a well-known marker of high-risk CLL, together with the expression of STAT1, NFATC2 and c-Myc, which represent downstream molecules of the B-cell proliferation/survival signaling pathway [50].

### 4.2. Programmed Cell Death 1 (PD-1)/Programmed Cell Death Ligand 1 (PD-L1)

Programmed cell death 1 (PD-1, also known as CD279) and its ligands programmed cell death ligand 1 (PD-L1, also known as B7-H1 and CD274) and PD-L2 (also known as B7-DC and CD273) are considered one of the most important axis in the maintenance of a tolerant microenvironment [51].

PD-1 is expressed on activated T cells upon TCR engagement, similarly to what described for CTLA-4 (Figure 2). However, at variance with the latter, PD-1 upregulation is not mediated by the rapid transport of the molecule at the cell surface, but requires transcriptional activation and it therefore occurs after few hours delay upon TCR stimulation. Functionally, upon binding to its ligand PD-L1, PD-1 clusters with TCR and recruits inhibitory phosphatases SHP-2 and PP2A. The cytoplasmic tail of PD-1 contains an immunoreceptor tyrosine-based inhibition motif (ITIM) and an immunoreceptor tyrosine-based switch motif (ITSM), both phosphorylated upon PD-1 stimulation and responsible for recruiting phosphatases at the cluster [52]. These negative co-stimulatory micro-clusters induce the dephosphorylation of the proximal TCR signaling molecules, thereby interfering with downstream activation and inducing an “exhausted” T-cell phenotype [53]. In cancer biology, PD-1 is upregulated on several immune cells including T, B and NK cells, where it exerts similar inhibitory effects. In contrast, PD-1 activation on Tregs [54] and myeloid-derived suppressor cells [55] can enhance their inhibitory function to further feed the impairment of T-cell mediated anti-tumor response.

Upregulation of PD-1 on different subpopulation of the CD4^+^ and the CD8^+^ T cell subsets is widely described in CLL, where it generally correlates with an inferior disease outcome and increased risk of infection, independently of other prognostic markers [56,57]. Compared with healthy donors, circulating T cells from CLL patients have been shown to have elevated PD-1 expression that is further upregulated upon in vitro T cell activation via CD3/CD28 [58,59]. Evidence of modulation of PD-1 expression with cell activation, comes also from the observation that in CLL lymph nodes the higher density of PD-1^+^ T cells is within the proliferation center, where CD4^+^ T lymphocytes are in close contact with activated leukemic B cells. Furthermore, in these microenvironmental areas reside CLL cells undergoing active proliferation and expansion that stain highly positive for PD-L1, suggesting a feed-forward loop of immune modulation [58]. PD-L1 is also upregulated on circulating CLL cells and its expression levels correlate between different disease compartments, being higher in lymph node and bone marrow. Similarly to what has been observed for PD-1 expression on T cells, no association was found between PD-L1 levels on leukemic cells and other disease prognosticators [60], although its expression was further upregulated upon in vitro activation of CLL cells with proliferative stimuli such as CpG/IL-2 [58]. PD-L1 is also expressed in the monocyte compartment, where it is upregulated in response to the transfer of CLL-derived exosomes, containing non-coding RNA, and through the activation of TLR-7 signaling [61]. Expression of the PD-1/PD-L1 axis on different cells of the CLL environment can be modulated by a number of transcriptional and non-transcriptional circuits. Demethylation of the distal upstream locus of the PD-1 gene *PDCD1* was described to occur upon activation of naïve CD8^+^ T cells by anti-CD3/anti-CD28 antibodies and IL-2, a mechanism shared with other genes encoding immunomodulatory markers [62]. More recently, Lewinsky and colleagues showed that PD-L1 expression on CLL cells is upregulated upon engagement of CD84, a surface molecule belonging to the SLAM family known to promote leukemic cell survival. The proposed mechanism of action involves the PI3K/AKT signaling cascade and the activation of the transcription factor STAT3. In addition, the authors showed that CD84 silencing not only downregulates PD-L1 expression on CLL cells and in the microenvironment, but it also affects PD-1 expression on the T cell compartment, suggesting a central role of CD84 in driving immunomodulatory signaling [63]. Lastly, treatment with Bruton’s tyrosine kinase (BTK) inhibitor ibrutinib was also found to decrease PD-1/PD-L1 expression, thereby affecting T cell proliferation and pseudoexhaustion, with important implication in the disease outcome [64,65,66,67,68,69].

From a functional standpoint, the effects and the impact of the PD-1/PD-L1 axis in CLL immune response dysregulation remains controversial. Riches and colleagues described impaired T-cell proliferation and cytotoxicity with maintained IFN-γ and TNF-α production, despite PD-1 expression [30]. Conversely, other reports highlighted that PD-1 expression is increased on proliferating compared with non-proliferating T cells and that its activation suppresses IFN-γ secretion by CD8^+^ T cells, while maintaining IL-4 secretion by CD4^+^ lymphocytes relatively unaffected and contributing to the overall Th2 skewing of T-cell responses observed in CLL patients [58]. In line with this view, PD-1 silencing was observed to restore physiological mechanisms of the immune synapse, enabling full activation of T lymphocytes [70]. The difficulties in determining a clear contribution of this immunomodulatory axis in CLL come from the fact that PD-1/PD-L1 interactions play a role in the senescence and aging of T cells even in physiological conditions, and this could represent a confounding factor considering that CLL is typically a disease of elderly people. However, PD-1^+^ normal T cells have markedly different effector functions compared to PD-1^+^ CLL T cells, the latter showing a significant impairment of cytotoxic abilities compared to normal T cells [71]. The study from McClanahan and colleagues argues in favor of a more complicated paradigm than the simple statement that PD-1^+^ CLL T cells are non-functional. By taking advantage of the TCL1 CLL mouse model [72], the authors show that PD-1^high^ T cells from leukemic mice have defects in IFN-γ and Granzyme B expression compared to aging normal mice, but they do not completely lose their cytotoxic capacity, although their ability to form immune synapses is significantly impaired compared with PD-1^low^ cells. The authors suggest that PD-1^+^ T cells in CLL are a functionally heterogeneous population in which key immunological nodes (i.e., recognition, generation of effector cells, effector potential) may be affected in different ways through multiple interactions occurring with leukemic cells in the microenvironment [71].

Over the last few years, strategies to target the PD-1/PD-L1 axis acquired growing relevance in the attempt to restore functional immune response in tumor biology. PD-1 antibodies currently approved by the US Food and Drug Administration (FDA) include nivolumab and pembrolizumab, two fully human or humanized monoclonal antibodies that block the binding of PD-L1/PD-L2 to PD-1 because of their high affinity and specificity. Functional assays demonstrated that treatment with nivolumab enhanced antigen-specific T-cell responses, including proliferation and cytokine production of effector T cells [73]. Combination trials are currently ongoing applied to B cell malignancies although, to date, their efficacy in CLL was proven limited compared to other lymphomas (NCT03514017, NCT02332980, NCT03884998; www.clinicaltrials.gov). Nivolumab combined with ibrutinib showed clinical activity in a small number of patients with relapsed/refractory CLL or CLL transforming into an aggressive lymphoma (Richter’s Transformation) [59,74]. Other preliminary studies showed an increased number of circulating long-lived central memory T cells with increased expression of co-signaling molecules (PD-L1^+^ lymphocytes) suggesting on-target in vivo effects of anti-PD-1 antibody therapy. Thus, targeting the PD-1/PD-L1 pathway has the potential to enhance anti-tumor immunity in B cell malignancies [75].

### 4.3. T Cell Immunoreceptor with Immunoglobulin and Immunoreceptor Tyrosine-Based Inhibition Motif (ITIM) Domains (TIGIT)

TIGIT (T cell immunoreceptor with Ig and ITIM domains) is an inhibitory receptor expressed on T, NK and NKT cells. It has been identified through a genomic search for genes specifically expressed in T cells and bearing a structure similar to other immunomodulatory receptors [76]. Despite being identified a decade ago, TIGIT reached the top visibility only recently, as witnessed by a significant number of papers in the last few years describing its involvement in suppressing immune response in various clinical conditions, including cancer. Nevertheless, its clinical significance in the CLL field is still a relatively unexplored topic. TIGIT shares structural and mechanistic similarities with PD-1 and CTLA-4, respectively. The cytoplasmic tail contains an immunoglobulin tail tyrosine (ITT)-like phosphorylation motif and an ITIM domain, through which TIGIT recruits the phosphatase SHIP1 and inhibits downstream activation of NF-κB, PI3K and MAPK pathways, similarly to what described for PD-1 [77,78]. TIGIT pairs with CD226/DNAM-1 (DNAX Accessory Molecule-1) in the sense that they compete with each other for the binding to the same set of ligands, resulting in opposite signaling outcomes and recalling the CTLA-4/CD28 dynamics (Figure 2) [79]. The two molecules partially share the expression pattern, although CD226 is more widely expressed on immune cells [80], whereas TIGIT is absent on naive T cells, but is expressed on activated and memory T cells, Tregs and on NK and NKT cells [76,81].

TIGIT and CD226 interact with the two nectin-family members poliovirus receptor (PVR) (CD155/Necl-5/Tage4) and poliovirus receptor-related 2 PVRL2 (CD112) [81], which are widely expressed on different cell types and on cancer cells, [82]. Notably, upregulation of CD155 surface levels was described on cancer cells, induced by Ras activation and genotoxic stress [83,84], but also on antigen-presenting cells (APCs) upon activation of TLR signaling [85].

Signaling triggered upon CD155 binding to CD226 potentiates CD8^+^ T cell and NK cell cytotoxicity toward tumor cells [86]. Upon engagement, CD226 is serine-phosphorylated and interacts with lymphocyte function-associated antigen 1 (LFA-1). This event induces subsequent CD226 tyrosine-phosphorylation and LFA-1-mediated recruitment to lipid rafts, providing co-stimulatory signaling that favor immune synapse formation and cytokine production by CD4^+^ and CD8^+^ T cells [87,88]. If TIGIT is concomitantly expressed on the cell surface, it prevents CD226 activation either by sequestering CD155 or by impeding CD226 homodimerization and phosphorylation [89]. Whether TIGIT triggers a full inhibitory cascade or functions by simply opposing the CD226-mediated positive co-stimulatory signal remains unclear. The observation that TIGIT engagement by an agonistic antibody decreases T cell activation when stimulated with anti-CD3 and anti-CD28 antibodies, and that its knockdown, conversely, increases T cell proliferation and effector cytokines production, suggest that a negative downstream signaling via TIGIT could arrest T cell activation [90]. However, several pieces of evidence argue in favor of a CD226-dependent TIGIT-mediated inhibition, at least when T cells are activated in the presence of CD155, as CD226 blockade abolishes the impact of TIGIT modulation on T cell proliferation and activity [89]. Consistently, upregulation of TIGIT expression is associated with an exhausted T cell phenotype [91]. Furthermore, aside from the inhibition exerted on T lymphocytes and NK cells, TIGIT promotes Tregs differentiation and enhances their suppressive activity [92].

Deregulation of the TIGIT/CD226/CD155 axis was described in several tumors: TIGIT^+^ tumor infiltrating CD8^+^ T cells could be detected in small-cell lung cancers, colorectal cancers and melanoma [93], and TIGIT inhibition was recently shown to increase T cell functions of melanoma specific CD8^+^ T cells [94]. Similar observations hold true also in CLL. TIGIT expression is upregulated in the CD4^+^ T cell compartment of CLL patients and is positively correlated with PD-1 expression in the same subset. TIGIT^+^/CD4^+^ T cells are enriched in high-risk CLL patients, as identified by advanced disease stage, unmutated IGHV genes or unfavorable cytogenetics. Functionally, TIGIT^+^/CD4^+^ T cells exhibit increased ability to sustain leukemic cell survival in co-culture experiments, and blocking TIGIT interactions using recombinant TIGIT-Fc molecules decreases CLL cells viability and interferes with production of prosurvival cytokines by CD4^+^ T cells [95].

Similar to other immunomodulatory molecules, therapeutic antibodies targeting TIGIT were developed and recently entered clinical trials limited to solid/metastatic tumors (NCT04047862, NCT03563716, NCT03628677, NCT02913313; www.clinicaltrials.gov). No trials are yet currently ongoing applied to CLL.

### 4.4. HLA-G

Surface expression of HLA-G may represent an additional mechanism through which tumor cells escape the immune response, although its relevance in CLL is still controversial. HLA-G belongs to the family of non-classical HLA class I antigens. This molecule differs from the classical HLA class I molecules with regard to variability in molecule structure, limited degree of polymorphism and immune functions. Physiologically, its expression is restricted to tissues where the immune system needs to be constantly suppressed, including the trophoblast, thymus, cornea, erythroid and endothelial precursors. HLA-G was first described as a ligand for inhibitory receptors present on uterine NK cells, thereby contributing to maternofetal tolerance [96]. Upregulation of this molecule in cancer can contribute to immunosuppression with deleterious implications. Tumor cells typically downregulate HLA-I to evade the T cell immune response and, in parallel, they variably upregulate HLA-G to abrogate, at least in part, compensatory NK cell cytotoxicity [97]. Multiple inhibitory functions of HLA-G were described against NK cells (inhibition of cytolysis, proliferation and migration), T lymphocytes (inhibition of cytotoxicity and proliferation), and APCs (impairment of dendritic cell maturation and antigen presentation) via direct binding to the inhibitory receptors ILT-2, ILT-4, and KIR2DL4 [98].

Increased plasma levels of soluble HLA-G (sHLA-G) have been reported in CLL patients compared to healthy donors [99,100], even though no significant correlation was found with known CLL prognosticators. Conversely, surface expression of HLA-G on leukemic cells was documented higher in patients with reduced treatment-free survival and came out as an independent predictor of disease progression in a multivariate Cox regression analysis, arguing in favor of its prognostic relevance in CLL (Figure 2) [101,102]. However, contradictory results were reported in a larger sample cohort where no correlation between surface HLA-G and treatment-free survival or other prognostic indicators, suggesting that the significance of HLA-G expression in CLL is yet far from being accredited [103].

Despite being a relatively low-polymorphic region, polymorphisms at the 5′ upstream regulatory and the 3′ untranslated regions of the *HLA-G* were described, with one of these, characterized by the deletion/insertion (del/ins) of 14 base pairs (14 bp) (rs66554220), found to be responsible for mRNA stability and consequently protein production [104]. In CLL, the presence of a del/del *HLA-G* genotype is associated with increased plasma levels of sHLA-G and only moderately increased surface expression, likely because membrane HLA-G forms are rather unstable and only transiently expressed on the cell surface to be rapidly released into the plasma. In addition, the del/del genotype was found to be associated with expansion of circulating Tregs, which in CLL positively correlates with the presence of clinical and biological features of aggressive disease. Furthermore, increased sHLA-G in del/del patients associates with impaired NK cytotoxicity through its binding to the KIR2DL4 ligand expressed by NK cells, as confirmed by in vitro incubation of normal NK cells with plasma samples from CLL patients with variable sHLA-G levels. Consistent with these findings, the del/del HLA-G genotype significantly associates with poorer overall survival compared to other genotypes (ins/del heterozygosity or ins/ins homozygosity) [105].

## 5. Environmental Mechanisms Driving Immune Response Dysfunction in CLL

### 5.1. The B Cell Receptor

The derailment of the B cell receptor (BCR) functions has a twofold outcome in CLL: on the one hand, it emerged as the driving force of the malignant behavior of leukemic cells, a topic that won’t be addressed here. On the other hand, its physiological functions in the establishment of a healthy immune function are almost completely lost in CLL, with strong implications in the scenario of immune response aberrations observed in the disease.

The BCR elicits some basic functions in B cell biology, including driving the maturation of long-lived functional B cells capable of rapidly responding to external pathogens by recognition of their individual cognate antigen. Secondly, it contributes to the development of antigen-presenting cell functions in mature B lymphocytes making them able to present antigen-derived peptides on the cell surface to helper CD4 T cells. Finally, it drives the production of high-affinity soluble immunoglobulins capable of uniquely recognizing and binding the cognate antigenic target [106]. Physiologically, the BCR maturation is a multistep process that generates a great diversity of ligand-binding regions through genetic rearrangements occurring early in pre–B-cell development, further implemented by somatic hypermutation and class-switch recombination, thus ensuring specific antibody production. However, in the majority of B cell malignancies, including CLL, the isotype of BCR-associated surface immunoglobulin (sIg) is restricted to IgM and somatic hypermutation may not have been occurred yet at the time of malignant transformation, marking leukemic cells with a less anergic profile and a more aggressive behavior. Furthermore, in CLL the BCR is often stereotyped, meaning that identical complementarity determining regions are found in CLL clones derived from different patients [107]. Stereotyped BCRs recognize similar antigenic structures, suggesting that continued antigenic stimulation, likely coming from auto-antigens, could be related to the development and progression of CLL. The global scenario of perturbation of the immune response further contributes to the expansion of auto-reactive clones thereby promoting autoimmune reactions in CLL [108]. In addition to a ligation-driven BCR signaling, antigen-independent cell-autonomous signaling, referred to as “tonic” signaling, is also present in CLL and witnessed by the increased phosphorylation of elements downstream to the BCR in the absence of antigens [109].

Deregulation of the BCR signaling is remarkably enhanced in the lymph node microenvironment, where antigens together with extra stimuli influencing CLL cells behavior are most abundant [110]. This is in line with the view that the lymphoid niche is the environment where CLL biology is shaped through multiple interactions between leukemic cells and non-tumoral bystander cells. This complex network includes both membrane-bound and soluble factors that contribute to the remodeling of the lymph node microenvironment towards a pro-leukemic immunotolerant milieu, where CLL cells receive stimulatory signals (e.g., aberrant activation of the BCR signaling) while directly orchestrating phenotypical and functional changes of immune response mediators.

### 5.2. Hypoxia and Metabolic Adaptation

The tumor microenvironment is typically a low oxygen milieu as a consequence of uncontrolled cancer cell proliferation and abnormal vascularization [111]. Hypoxia is progressively established as a consequence of increased oxygen consumption by hyperproliferating cancer cells and, in turn, boosts a glycolysis-mediated energy supply, required for proliferation, via the activation of HIF1-α-driven transcriptional reprogramming [112]. This feed-forward loop leads to the acidification of the extracellular space, due the accumulation of extracellular lactate [113], which was described to affect tumor-associated immune cells and stromal elements leading to a more tolerant environment. Increased lactate attenuates the activation of dendritic cells and T cells, inhibits monocytes migration and favors macrophages polarization towards an M2-phenotype [114,115,116,117].

Similarly to other tumor environments, the lymphoid niche is highly hypoxic [118]. CLL cells and bystanders non-leukemic cells undergo metabolic adaptation upon oxygen deprivation, rapidly fostering energy production via glycolysis through HIF1-α-mediated transcriptional control and robustly upregulating glycolytic enzymes and glucose and lactate transporters [118,119]. Hypoxia-driven immunomodulation was described to occur in CLL microenvironment. Leukemic cells cultured under low oxygen tension benefit from increased survival and cytoprotection and upregulate IL-10 production, enhancing their B-regulatory phenotype. Likewise, metabolic switch toward glycolysis skews T cells functions to a regulatory phenotype by upregulating the expression of FOXP3 [120], together with PD-1, IL-10, and VEGFA, while reducing IFN-γ [118].

Moreover, metabolic adaptation is accompanied by increased generation and release in the extracellular space of intermediate products and co-factors such as nicotinamide adenine dinucleotide (NAD^+^) and adenosine triphosphate (ATP) [121] that can be further metabolized to adenosine (discussed in the next paragraph) [122]. In particular, NAD^+^ is critical in a number of metabolic and biological processes, being used as a redox coenzyme by several dehydrogenases, and as a co-substrate by various NAD-consuming enzymes, such as mono- or poly-ADP ribosyltransferases and sirtuins [123]. Coenzyme regeneration is guaranteed through different biosynthetic routes, among which the one catalyzed by NAMPT is considered the major pathway ensuring NAD^+^ homeostasis [124]. Aside from its role as NAD-biosynthetic enzyme, NAMPT can be released in the extracellular space (eNAMPT) where it acts as a cytokine with immune cell modulation properties. eNAMPT, also known as pre-B-cell colony enhancing factor PBEF, was originally described to be secreted by activated lymphocytes and bone marrow stromal cells [125]. More recently, increased amount of eNAMPT were observed in plasma samples from CLL patients resulting from upregulated production and release in the extracellular space by activated CLL cells. Activation of CLL lymphocytes through the stimulation of the BCR, the TLR-9 and the CD40 signaling pathway, converging on NF-κB activation, was shown to induce eNAMPT increase. Once in the extracellular space, eNAMPT promotes differentiation of resting monocytes, polarizing them toward tumor-supporting M2 macrophages, which express high levels of CD163, CD206, and indoleamine 2,3-dioxygenase (IDO) and secrete immunosuppressive cytokines [126]. eNAMPT exerts its effects through binding to TLR-4 and inducing STAT3 and NF-κB signaling (Figure 2) [127]. As a consequence of phenotypic and functional skewing, eNAMPT-primed macrophages strongly support CLL cells’ survival while reducing T cell responses [126].

### 5.3. Adenosine Signaling

Increased levels of ATP and NAD^+^ in the extracellular space can be found during inflammatory, hypoxic or neoplastic conditions, where they are believed to act as danger signals, alerting the immune system to possible tissue damage [121]. Extracellular homeostasis is restored through a scavenging circuit operated by nucleotide-catabolizing enzymes that produce the immunosuppressant adenosine (ADO) and inosine, which can re-enter the cell, reconstituting the nucleotide pool [128]. ATP is degraded through the sequential action of CD39 (ectonucleoside triphosphate diphosphohydrolase-1, ENTPD-1), which converts extracellular ATP (or ADP) to adenosine monophosphate (AMP), and CD73 (5’-nucleotidase) that converts AMP to ADO. Alternatively, ADO can be generated from NAD^+^ through the coordinated action of CD38 (NAD glycohydrolase), which generates ADP ribose, and PC-1 (ectonucleotide pyrophosphatase/phosphodiesterase family member 1), which generates AMP. However coming from ATP or NAD^+^ dismantling, AMP is degraded to ADO by CD73, which consequently acts as bottleneck enzyme for both cascades and is considered the rate-limiting enzyme in ADO generation [129]. Aside from being metabolized to inosine by the adenosine deaminase (ADA), extracellular ADO can bind to one of four type 1 (P1) purinergic adenosine receptors (A1, A2A, A2B, and A3), G-protein-coupled receptors that signal through the cyclic AMP (cAMP) and the MAPK pathway [130,131].

ADO signaling is widely associated with immunosuppression in cancer [132]. Likewise, marked upregulation of surface CD39, CD73 and A2A expression have been described on CLL cells, further enhanced in the presence of activating stimuli, and associated to a more aggressive clinical behavior and an intense leukemic cell proliferation [133]. The expression and function of the adenosinergic axis are boosted under hypoxic conditions [134]. Consistently, CLL cells collected from lymph nodes express significantly higher levels of CD73, A2A and nucleoside transporters and this holds true also for macrophages and T lymphocytes residing in this environment [118]. Strong evidence argues in favor of adenosine mediating at least in part hypoxia-driven immunosuppression in the CLL microenvironment, with the re-shaping of a tumor-favorable niche. Specifically, treatment of macrophages with an A2A agonist recapitulates the hypoxia-driven polarization towards an M2 phenotype, with upregulation of markers such as the transcription factor IRF4 and the tryptophan-metabolizing enzyme IDO, and enhances the growth-supportive ability of NLCs by increasing synthesis of IL-6, which confers growth advantage to CLL cells. Similarly, ADO signaling significantly impairs T cell proliferation in response to activating stimuli, such as anti-CD3/CD28 ligation (Figure 2). The proof of principle that adenosine is one of the main effectors of hypoxia-driven circuits comes from the observation that the A2A inhibitor SCH58261 rescues the immunosuppressive phenotype and functions of B cells, macrophages and T cells cultured under low oxygen tension. For example, SCH58261 leads to a significant decrease of IL-10 and IL-6 production, to the downmodulation of M2 and T-cell exhaustion markers, and to reduced Tregs expansion [118].

## 6. Concluding Remarks

A broad range of defects of the innate and adaptive immune response is found in CLL patients since early disease stage. Over the years, several approaches aimed at evoking the immune response against leukemia have been developed, although none of them so far appeared to be as revolutionary and successful in CLL as some of them are in other tumor models, including lymphomas [135,136]. This is the case, for example, of the PD-1 blocking antibodies that did not show any evidence of clinical response or benefit in CLL trials, neither as single agents nor in combination with ibrutinib, unless patients underwent Richter’s transformation [137,138]. While the reason for this lack of response remains to be determined, impairment in complement activity and reduced levels of complement proteins may contribute to limiting the efficacy of monoclonal antibodies, which rely on complement-dependent cytotoxicity [139]. On the other hand, anti-CD20 monoclonal antibodies have shown significant responses in combination with chemotherapeutic or targeted agents [140], suggesting that the picture is probably more complex.

By far the most innovative immunotherapeutic strategy developed in the last decade has been the use of chimeric antigen receptor (CAR)-transduced T cells. CAR constructs are designed to simultaneously redirect T cells toward tumor cells while inducing activating signaling and effector responses. This revolutionary approach uses autologous T cells from patients, thereby excluding graft-versus-host disease complications, and, if successful, it leads to the establishment of memory T cells, inducing prolonged protection against the tumor [141]. Despite the high response rate achieved by CAR-T cells in other hematological malignancies, the rate of complete remission in CLL is currently far lower, although at least a large proportion of patients showed a clinical response to CD19 CAR-T cells [142]. The challenge now is to ameliorate and to increase the rate of complete remissions, given also that patients who achieve it have very low risk of disease relapse [143].

The explanation behind the overall disappointing results obtained with monoclonal antibodies or with cell cytotoxicity-based approaches in CLL (reviewed in [144]) is believed to reside in the profound alterations of different aspects of the immune system and in the pivotal role played by leukemic cells themselves in deranging the immune response. In other words, in CLL there is not a dominant immunosuppressive mechanism that can be appointed as a key target to restore immune competence, but a complicated and redundant network of molecular and metabolic axes that can compensate each other’s functions when targeted by immunotherapeutic strategies.

## Figures and Tables

**Figure 1 ijms-21-01825-f001:**
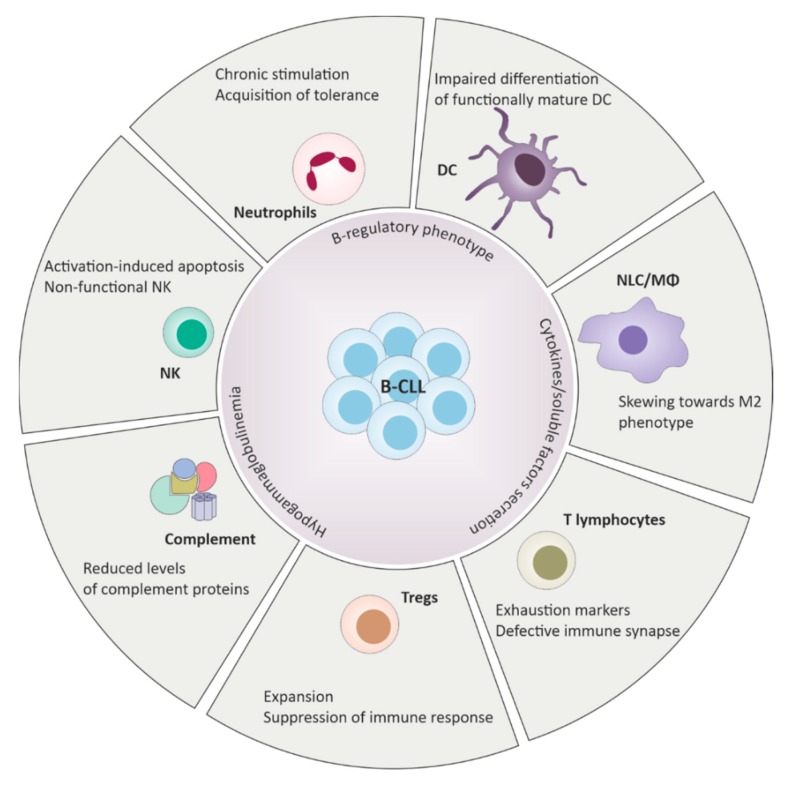
Defects of the innate and the adaptive immune response in chronic lymphocytic leukemia (CLL). B-CLL cells harbor B-regulatory properties and play a central role in driving immunosuppression through cell-cell contacts and cytokines release. Deregulation of the immune system includes wide phenotypical changes and functional defects across different elements of the immune surveillance, ultimately promoting tolerance and favoring tumor expansion.

**Figure 2 ijms-21-01825-f002:**
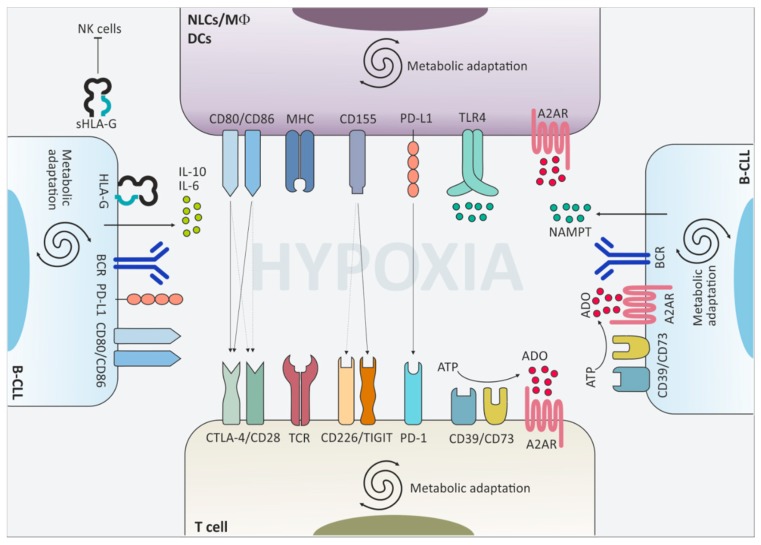
Surface molecules and extracellular mediators contribute to the downregulation of immune cell functions. The CLL niche is an hypoxic environment where leukemic cells interact with non-tumor cells and shape their phenotype and function towards tolerance. Upregulation of cytotoxic T lymphocyte antigen-4 (CTLA-4), T cell immunoreceptor with immunoglobulin and immunoreceptor tyrosine-based inhibition motif domains (TIGIT) and programmed cell death 1 (PD-1) on the cell surface of T lymphocytes downmodulates T cell receptor (TCR)-mediated signaling and limits T cell activation. This effect is fostered also by the upregulation of the respective partner proteins on macrophages, dendritic cells and leukemic cells themselves. Moreover, CLL cells secrete pro-tolerant cytokines such as IL-10 and IL-6 and release nicotinamide phosphorybosyltransferase (NAMPT) in the extracellular space, which promote skewing of macrophages towards an M2-like phenotype. Upregulation of the adenosine-generating and -sensing molecules further boosts immunomodulation in CLL.

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
