# Peer review of "Immune Response Dysfunction in Chronic Lymphocytic Leukemia: Dissecting Molecular Mechanisms and Microenvironmental Conditions"

_ijms, 2020, doi:10.3390/ijms21051825_

Round 1

Reviewer 1 Report

This is a fine review summarizing immune disfunction in CLL. The review is well written focusing on innate and adaptive response in CLL on one side, while highlighting the role of major molecular pathways on the other.

Author Response

We thank the reviewer for the positive evaluation of our work.

Reviewer 2 Report

General impression:

Well written overview of defects in immune check points and hypoxia-driven changes in CLL.

The review provides detailed descriptions at the molecular level of regulatory and modulatory mechanisms that underlie such immune-phenotypical and functional alterations seen in the CLL microenvironment. However, the focus of the review does not seem to reflect the review title that aims at covering much broader perspectives of immune dysfunction in CLL. Furthermore, the review seems somehow biased towards the research interest of the authors rather than the broad field of immune dysfunction in CLL, most clearly reflected in the sections on HLA-G and hypoxia, which almost site work by the authors for references related to CLL.

Major issues:

The focus of the review as outlined in the title and the introduction is not reflected in the main text (focusing only on immune check points and hypoxia-induced changes). Cytokines and more general immune dysfunction mechanisms as well as the interplay between the B cell receptor pathway and the microenvironment is almost absent from the main part of the review and the dicussion

Mainly non-CLL related references are included for most of the text, in particular for TIGIT, HLA-G and hypoxia sections, for the two latter the only CLL-related citations are from the authors.

The clinical focus of immune dysregulation in CLL patients, including infections and secondary malignancies, is quickly lost in the review.

The review in general should emphasize that no clinical success with check point inhibitors have been achieved in CLL, this is only stated in a few places and almost ignored in other parts of the manuscript

In Concluding remarks, a whole new discussion of lack of effect of immunotherapy in CLL is opened. This part contradicts the clinical findings of FCR treatment and CD20-chlorambucil treatment improving overall survival in CLL for the first time in clinical trials compared to chemotherapy alone. Furthermore, despite the immune dysfunction in CLL, actually the first published CAR-T cell therapy was in CLL.   L 501-504 is misleading and should be omitted or fully refocused, see CLL11, CLL8, CLL10 and CLL14 clinical data.

Other comments:

1.Introduction

Too general, lacking presenting the significant data on specific immune dysfunction in CLL 33: consider mentioning autoimmunity among clinical manifestations of immune dysfunction in CLL. A quite long list of clinical factors affecting infectious risk in CLL – could be shortened Missing in the introduction is a general overview of the importance of the microenvironment for CLL cells – how CLL cells depend on microenvironmental survival signals, the importance of the b-cell receptor signaling in CLL-microenvironment interactions etc., as it helps create a general understanding of the relevance of the rest of the paper Citations for the introduction is mainly other reviews, not original work

Innate Immune Response Dysfunction in CLL Clarify that NLCs are an in vitro phenomenon and specify that the M2 skewed macrophages and TAMs in vivo resemble in vitro NLCs Monocytes and macrophages are used as synonyms, consider to distinguish these cell types as they can be functionally and phenotypically very different in vivo (in patients), and again clarify when describing in vitro phenomena Consider including myeloid derived suppressor cells under myeloid cells, as they probably play an important role in the suppressive milieu in CLL: Jitschin et al, Blood, 2014, Gustafson et al, BJH, 2012

Adaptive Immune Response Dysfunction in CLL Again, it needs to be clarified when in vitro / in vivo / in patient. Consider to include a clinical perspective of increased T-cells and Tregs, including effects of ibrutinib, chemoimmunotherapy and venetoclax on these cell subtypes

Molecular mechanisms driving immune response dysfunction in CLL Figure 2 (and 1) lack references in the figure legend and/or text detailing the work between every cell/molecule/mechanism mentioned 172: Clarify that we’re talking about CTLA expression on T-cells. 240: Here, MDSCs are mentioned – I suggest to describe them under myeloid cells in segment 2. Correlations between PD-1/PD-1L upregulation and infectious risk? 259-271: Clarify which mechanisms that regulate the PD-1/PD-L1 axis UP vs DOWN, rather than just using “modify” Effect of Ibrutinib on T-cell PD-1 expression and PD-1 expression in CLL in general, would benefit from additional citations/detailing: Niemann et al, CCR, 2016, Dubovsky et al, Blood 2013, Sagiv-Barfi et al., PNAS, 2015, Palma et al., Haematologica 2016 302: Consider to discuss here why PD-1 blockade is not as effective in CLL as in DLBCL, Richter and other lymphomas, and why ibrutinib potentially may improve response to PD-1 inhibition in CLL, given that ibrutinib treatment seems to down-regulate PD-1 on T-cells TIGIT and HLA-G: The relevance in CLL is mentioned quite late in these segments – consider to include something about CLL also starting the segments. This reviewer is not convinced of the importance to include these two areas for the review at all. TIGIT is presented as novel; more than 10 years of publications on this may not be novel.

Environmental mechanisms driving immune response dysfunction in CLL The whole section is reflecting a single aspect (hypoxia-driven changes) but omitting the major issue in terms of microenvironmental interactions leading to immune dysfunction in CLL due to cytokine/chemokine changes and cellular interactions. Hypoxia: It could be of interest to include the aspect of increased oxidative phosphorylation as a mechanism of resistance to venetoclax, that can be reversed through oxphos-inhibition: Chen et al, Abstract 479, ASH 2019

Author Response

Dear Reviewer #2,

Thanks for the overall positive evaluation of our work and for the constructive criticisms.

Below please find a point-by-point reply to the issues raised.

Major issues:

Reviewer’s issue: The focus of the review as outlined in the title and the introduction is not reflected in the main text (focusing only on immune check points and hypoxia-induced changes). Cytokines and more general immune dysfunction mechanisms as well as the interplay between the B cell receptor pathway and the microenvironment is almost absent from the main part of the review and the dicussion

Author’s reply: Following your suggestions, we have significantly modified the text, by revisiting and reorganizing the introduction, the part on myeloid cells and innate immunity, as well as by adding a whole new paragraph on the B cell receptor signaling and role in this context. We have decided not to add a dedicated paragraph for cytokines because we felt it would be repetitive as most of the information is interspersed in the text.

Reviewer’s issue: Mainly non-CLL related references are included for most of the text, in particular for TIGIT, HLA-G and hypoxia sections, for the two latter the only CLL-related citations are from the authors.

Author’s reply: We agree on this and the reason is that there limited literature on TIGIT and CLL and on the role of tissue hypoxia in the disease. Data on HLA-G is more numerous and is included in the text. 

Reviewer’s issue: The clinical focus of immune dysregulation in CLL patients, including infections and secondary malignancies, is quickly lost in the review.

Author’s reply: We have modified the introduction, trying to underline the clinical significance of immunosuppression. However, this is not a clinical review, but mostly deals with reports on molecular mechanisms of host adaptation to CLL

Reviewer’s issue: The review in general should emphasize that no clinical success with check point inhibitors have been achieved in CLL, this is only stated in a few places and almost ignored in other parts of the manuscript

Author’s reply: Thanks for pointing this out: it has been amended in the revised version.

Reviewer’s issue: In Concluding remarks, a whole new discussion of lack of effect of immunotherapy in CLL is opened. This part contradicts the clinical findings of FCR treatment and CD20-chlorambucil treatment improving overall survival in CLL for the first time in clinical trials compared to chemotherapy alone. Furthermore, despite the immune dysfunction in CLL, actually the first published CAR-T cell therapy was in CLL.   L 501-504 is misleading and should be omitted or fully refocused, see CLL11, CLL8, CLL10 and CLL14 clinical data.  

Author’s reply: We agree with the reviewer that that part was misleading. We have re-written the paragraph to improve readability and clarity.

Reviewer’s issue: Too general, lacking presenting the significant data on specific immune dysfunction in CLL 33: consider mentioning autoimmunity among clinical manifestations of immune dysfunction in CLL. A quite long list of clinical factors affecting infectious risk in CLL – could be shortened Missing in the introduction is a general overview of the importance of the microenvironment for CLL cells – how CLL cells depend on microenvironmental survival signals, the importance of the b-cell receptor signaling in CLL-microenvironment interactions etc., as it helps create a general understanding of the relevance of the rest of the paper Citations for the introduction is mainly other reviews, not original work

Author’s reply: Introduction was reshaped following your suggestions. Modifications are in red: we believe it now reads smoothly.

Reviewer’s issue:  Innate Immune Response Dysfunction in CLL Clarify that NLCs are an in vitro phenomenon and specify that the M2 skewed macrophages and TAMs in vivo resemble in vitro NLCs Monocytes and macrophages are used as synonyms, consider to distinguish these cell types as they can be functionally and phenotypically very different in vivo (in patients), and again clarify when describing in vitro phenomena Consider including myeloid derived suppressor cells under myeloid cells, as they probably play an important role in the suppressive milieu in CLL: Jitschin et al, Blood, 2014, Gustafson et al, BJH, 2012

Author’s reply: This part of the review has been modified as per your suggestion. Relevant references added.

Reviewer’s issue:  Adaptive Immune Response Dysfunction in CLL Again, it needs to be clarified when in vitro / in vivo / in patient. Consider to include a clinical perspective of increased T-cells and Tregs, including effects of ibrutinib, chemoimmunotherapy and venetoclax on these cell subtypes

Author’s reply:  Thanks for pointing this out: we discussed it , but decided against adding a paragraph with a clinical perspective, as it would add significant complexity, with the result of making the review too long and unfocused. 

Reviewer’s issue:  Molecular mechanisms driving immune response dysfunction in CLL Figure 2 (and 1) lack references in the figure legend and/or text detailing the work between every cell/molecule/mechanism mentioned 172: Clarify that we’re talking about CTLA expression on T-cells. 240: Here, MDSCs are mentioned – I suggest to describe them under myeloid cells in segment 2. Correlations between PD-1/PD-1L upregulation and infectious risk? 259-271: Clarify which mechanisms that regulate the PD-1/PD-L1 axis UP vs DOWN, rather than just using “modify” Effect of Ibrutinib on T-cell PD-1 expression and PD-1 expression in CLL in general, would benefit from additional citations/detailing: Niemann et al, CCR, 2016, Dubovsky et al, Blood 2013, Sagiv-Barfi et al., PNAS, 2015, Palma et al., Haematologica 2016 302: Consider to discuss here why PD-1 blockade is not as effective in CLL as in DLBCL, Richter and other lymphomas, and why ibrutinib potentially may improve response to PD-1 inhibition in CLL, given that ibrutinib treatment seems to down-regulate PD-1 on T-cells TIGIT and HLA-G: The relevance in CLL is mentioned quite late in these segments – consider to include something about CLL also starting the segments. This reviewer is not convinced of the importance to include these two areas for the review at all. TIGIT is presented as novel; more than 10 years of publications on this may not be novel.

Author’s reply: Thanks for these suggestions, they were mostly included. We would like to keep TIGIT in the discussion as it may be a potential new area. Suggested references are now included.

Reviewer’s issue: Environmental mechanisms driving immune response dysfunction in CLL The whole section is reflecting a single aspect (hypoxia-driven changes) but omitting the major issue in terms of microenvironmental interactions leading to immune dysfunction in CLL due to cytokine/chemokine changes and cellular interactions. Hypoxia: It could be of interest to include the aspect of increased oxidative phosphorylation as a mechanism of resistance to venetoclax, that can be reversed through oxphos-inhibition: Chen et al, Abstract 479, ASH 2019

Author’s reply: Thanks for pointing this out. It is true that to some extent this review represents the views of the authors, as well as our scientific background and interests. For these reasons, we would like to keep this part as is, without modifying its organization. 

Round 2

Reviewer 2 Report

Thank you for the responses to comments.

From the perspective of this reviewer, the paper still seems somehow biased in terms of the focus, thus I suggest to amend the title and abstract to reflect that the review focus on less commonly reported aspects of the microenvironment in CLL.

Author Response

Dear Reviewer #2,

I agree with you that this review provides a personal interpretation of the current status of the field, reflecting our views, personal interests and in a way our  "tastes".  However, it also summarizes the main topics of the field. In your first review, you listed some of the aspects which you felt were poorly covered. Following your recommendations, for which we thank you, we revised the text trying to cover all aspects of this very complex and still somewhat unknown field. As you can see from the revised version, we added specific paragraphs, modified the introduction and made the reference list more complete.

So I don't agree that this review "focuses on less commonly reported aspects of the microenvironment of CLL". It tries to touch on the majority of known aspects, while underlining our "favorites". I believe this is what a review should do.

In conclusion, I don't see any reason for changing the title or the abstract.

With kind regards,

Silvia Deaglio, MD, PhD